# Neuroprotective Effects of Neuropeptide Y on Human Neuroblastoma SH-SY5Y Cells in Glutamate Excitotoxicity and ER Stress Conditions

**DOI:** 10.3390/cells11223665

**Published:** 2022-11-18

**Authors:** Viswanthram Palanivel, Vivek Gupta, Seyed Shahab Oddin Mirshahvaladi, Samridhi Sharma, Veer Gupta, Nitin Chitranshi, Mehdi Mirzaei, Stuart L Graham, Devaraj Basavarajappa

**Affiliations:** 1Macquarie Medical School, Faculty of Medicine, Health and Human Sciences, Macquarie University, North Ryde, Sydney, NSW 2109, Australia; 2School of Medicine, Deakin University, Geelong, VIC 3216, Australia; 3Save Sight Institute, The University of Sydney, Sydney, NSW 2000, Australia

**Keywords:** neuropeptide Y, glutamate excitotoxicity, tunicamycin-induced ER stress, oxidative stress, neurodegeneration, neuroprotection, SH-SY5Y cells

## Abstract

Neuropeptide Y (NPY), a sympathetic neurotransmitter, is involved in various physiological functions, and its dysregulation is implicated in several neurodegenerative diseases. Glutamate excitotoxicity, endoplasmic reticulum (ER) stress, and oxidative stress are the common mechanisms associated with numerous neurodegenerative illnesses. The present study aimed to elucidate the protective effects of NPY against glutamate toxicity and tunicamycin-induced ER stress in the human neuroblastoma SH-SY5Y cell line. We exposed the SH-SY5Y cells to glutamate and tunicamycin for two different time points and analyzed the protective effects of NPY at different concentrations. The protective effects of NPY treatments were assessed by cell viability assay, and the signalling pathway changes were evaluated by biochemical techniques such as Western blotting and immunofluorescence assays. Our results showed that treatment of SH-SY5Y cells with NPY significantly increased the viability of the cells in both glutamate toxicity and ER stress conditions. NPY treatments significantly attenuated the glutamate-induced pro-apoptotic activation of ERK1/2 and JNK/BAD pathways. The protective effects of NPY were further evident against tunicamycin-induced ER stress. NPY treatments significantly suppressed the ER stress activation by downregulating BiP, phospho-eIF2α, and CHOP expression. In addition, NPY alleviated the Akt/FoxO3a pathway in acute oxidative conditions caused by glutamate and tunicamycin in SH-SY5Y cells. Our results demonstrated that NPY is neuroprotective against glutamate-induced cell toxicity and tunicamycin-induced ER stress through anti-apoptotic actions.

## 1. Introduction

Amongst different neuronal injuries, glutamate-induced excitotoxicity, endoplasmic reticulum (ER) stress, and oxidative stress are the common pathological mechanisms that are associated with several neurodegenerative diseases such as Alzheimer’s disease (AD), Parkinson’s disease (PD), Huntington’s disease (HD), epilepsy, ischemic stroke, glaucoma, and age-associated macular degeneration [1,2,3,4,5,6]. Glutamate is a primary excitatory neurotransmitter and plays a crucial physiological role in neural transmission, differentiation, and synaptic plasticity in the central nervous system (CNS) [2]. Excessive accumulation and impaired glutamate uptake in the nervous tissue lead to neuronal dysfunction and death [7]. The hyperexcitability of neurons caused by the excessive glutamate leads to calcium overload, increased intracellular reactive oxygen free radicals, mitochondrial dysfunction, DNA damage, and eventually apoptosis [7,8]. 

ER is a crucial intracellular organelle in higher eukaryotic cells involved in processing newly synthesized proteins for proper folding, trafficking, quality check, and degradation [9]. ER dysfunction leads to the accumulation of misfolded proteins, causing ER stress. ER stress triggers an adaptive signalling reaction known as the unfolded protein response (UPR) and promotes quality control mechanisms [3]. The UPR is mediated via the activation of PKR-like ER kinase (PERK), inositol-requiring transmembrane kinase/endoribonuclease 1α (IRE1α), and activating transcription factor 6 (ATF6) pathways [10]. However, dysfunctioning and prolonged activation of UPR play an essential role in promoting pro-apoptotic events [11,12]. Chronic ER stress has been identified closely with neuronal damage, and the accumulation of pathological protein aggregates within the nervous tissue is well documented in many neurodegenerative diseases [3,13,14]. Protein aggregates composed of misfolded proteins such as amyloid β (Aβ) or tau protein in AD, α-synuclein (α-syn) in PD, and mutant huntingtin protein (mHtt) in HD are the key hallmarks in these disorders, suggesting that ER stress has a vital role in the pathophysiology of these neurodegenerative diseases [13,15]. 

Neuropeptide Y (NPY) is an endogenous 36-amino acid linear peptide with tyrosine moieties on both ends of the molecule [16]. NPY is widely expressed in the CNS and peripheral nervous systems (PNS) in mammals and contributes to a vast number of physiological processes, including regulation of brain activity, appetite stimuli, stress responses, blood pressure, heart rate homeostasis, body metabolism, and immune function regulation [17,18]. It belongs to a family of neuroendocrine peptides, which also includes peptide YY (PYY), and pancreatic polypeptides (PP), and the members of this family share a secondary structure termed the ‘PP-fold’ comprising extended polyproline helix and an α-helix connected by a β-turn with tyrosine residues at both the ends [16]. NPY binds and exerts its effects through NPY receptors, which are class A or rhodopsin-like G-protein coupled receptors (GPCRs) with different isoforms [19]. Four NPY receptors such as hY1R, hY2R, hY4R, and hY5R have been reported to be functional in humans [17,20]. 

NPY was initially speculated to be only a neurotransmitter; however, further research in human and animal models has revealed its diverse pathophysiological functions in many neurodegenerative diseases [18,21]. As a neuromodulator, NPY influences the release of several neurotransmitters and modulates calcium influx and neurogenesis [22]. In the CNS, the NPY receptors activated by NPY induce neuroprotective effects, including excitotoxicity reduction, regulation of calcium homeostasis, stimulation of autophagy, attenuating neuroinflammation, promotion of neurogenesis, and increasing trophic support [22,23,24,25,26,27]. Further, unravelling NPY receptor subtypes has provided numerous opportunities to understand their functions. The neuronal proliferation induced by NPY occurs via ERK1/2 and Akt pathways, inhibiting necrotic and apoptotic cell death in neurodegenerative diseases [23,28]. Cumulative evidence suggests that NPY and its receptors in the CNS may be potential neuroprotective targets in various degenerative conditions [26,29,30]. In this study, we have investigated the protective role of NPY against glutamate toxicity and ER stress in human neuroblastoma SH-SY5Y cells and determined its role in downregulating cellular pro-apoptotic events.

## 2. Materials and Methods

### 2.1. Cell Culture

The human neuroblastoma SH-SY5Y cell line was purchased from the American Type Culture Collection (ATCC, Manassas, VA, USA, RRID:CVCL_0019). The cells were cultured in a Dulbecco’s modified eagle medium (DMEM; Thermo Fisher, Waltham, MA, USA) with L-glutamine, L-glucose, sodium pyruvate and supplemented with 10% fetal bovine serum (FBS; Thermo Fisher, Waltham, MA, USA), and the antibiotics, 100 units/mL penicillin and 100 µg/mL streptomycin (Thermo Fisher, Waltham, MA, USA). The cells were maintained at 37 °C in a humidified atmosphere with 5% CO_2_. Trypsin-EDTA (Sigma-Aldrich, Darmstadt, Germany) solution was used for passaging the cells

### 2.2. Cell Treatments with Glutamate, Tunicamycin, and Neuropeptide Y

L-Glutamic acid (Sigma-Aldrich, Darmstadt, Germany) was dissolved at 25 °C in a complete culture medium to a stock concentration of 100 mM and then adjusted to pH 7.5 using 1M NaOH. A freshly prepared L-glutamate was used in all experiments, and it was further dissolved in a culture medium to obtain the required concentrations. Antibiotic tunicamycin is considered as a rapid ER stress inducer and is widely used in many in vitro studies [31,32]. Tunicamycin (Tocris Bioscience, Abingdon, UK) was dissolved in dimethyl sulfoxide (DMSO; Sigma, Saint Louis, MO, USA) to a concentration of 1µg/mL, and it was further dissolved in a culture medium to obtain appropriate working concentrations. Neuropeptide Y (endogenous peptide ab120208; Abcam, Cambridge, UK) was dissolved in sterile distilled water at a concentration of 100 µM, stored as aliquots at −80 °C, and thawed once before treatments. All the treatments were carried out when cells reached about 60% confluency. A wide range of L-glutamate (0–80 mM) and tunicamycin (0–8 µg/mL) concentrations were selected based on concentrations used in previous studies to test the cell viability [32,33]. Similarly, protective effects of NPY at different concentrations were determined based on the efficiency of the peptide described in previous in vitro studies [34,35]. 

For analyzing the toxic effects of L-glutamate or tunicamycin at different concentrations, the cells were treated with 20, 40, 60, and 80 mM of L-glutamate or with 1, 3, 6, and 8 µg/mL of tunicamycin for 12 and 24 h. To evaluate the protective effects of NPY against L-glutamate and tunicamycin toxicity, the cells were grouped as (1) a control group with no treatment (untreated); (2) NPY treated group where cells were treated with 0.1 µM, 0.2 µM, 0.5 µM, and 1 µM of NPY; (3) L-glutamate (40 mM) or tunicamycin (1 µg/mL) treated group and (4) toxicity + NPY treated group where cells were co-treated with L-glutamate (40 mM) or tunicamycin (1 µg/mL) and 0.1 µM, 0.2 µM, 0.5 µM, and 1 µM of NPY, respectively.

### 2.3. Cell Viability Test 

Cell viability was performed using a 3-(4,5-Dimethylthiazol-2-yl)-2,5-diphenyltetrazolium bromide (MTT; Sigma, Saint Louis, MO, USA) assay [36,37]. Briefly, SH-SY5 Y cells were seeded into 96 well plates at a density of 7 × 10^3^ cells per well and incubated for 24 h for attachment before the treatments, as mentioned earlier. Assays were performed 12 and 24 h after the treatments to determine cell viability. MTT was added to each well at a 12 mM concentration and incubated for 4 h at 37 °C. After incubation, the MTT solution was removed from the wells, and 100 µL of DMSO was added to solubilize the purple formazan crystals. After an hour of incubation, the absorbance was measured at 532 nm and 650 nm using a microplate reader (CLARIOstar Plus, BMG Labtech, Ortenberg, Germany). The untreated control cell group was considered as 100% cell viability. 

### 2.4. SDS-PAGE and Western Blotting 

The expression of proteins in treated SH-SY5Y cells was assessed by Western blotting. SH-SY5Y cells were seeded into 6 well plates at a density of 1 × 10^5^ cells per well. Cells were harvested after 12 and 24 h of different treatments. The harvested cells were then lysed in ice-cold RIPA lysis buffer (50 mM Tris, pH-8.0, 150 mM NaCl, 1% Triton-X-100, 0.5%) supplemented with 1% phosphatase inhibitor (PhosSTOP, Sigma-Aldrich, Saint Louis, MO, USA) and 1% protease inhibitor cocktail (Sigma-Aldrich, Saint Louis, MO, USA). The cell suspensions with RIPA lysis buffer were sonicated and centrifuged at 12,000× *g* for 15 min at 4 °C to pellet down the cell debris. The whole cell lysate supernatant was collected and used as the total protein extract. The concentration of the protein extracts was quantified by the Pierce BCA assay kit (Thermo Fisher Scientific, Waltham, MA, USA). Purified BSA (0–2000 µg/mL) was used as the standard to generate the linear response curve. All samples were quantified as triplicates in a 96-well plate. The absorbance was measured at 562 nm using a microplate reader and the unknown protein concentrations were interpolated from the BSA standard curve. Proteins (precisely 10 µg) were subjected to SDS-PAGE on 4–12% gels and transferred to nitrocellulose membranes by electroblotting (Invitrogen iBlot2, Thermo Fisher Scientific, Waltham, MA, USA) as described previously [38,39,40]. Following protein transfer, the membranes were blocked with non-fat dry skim milk powder (5%) in TTBS (20 mM Tris-HCl pH 7.6, 100 mM NaCl, 0.1% Tween 20) for 1 h at room temperature. The blots were then incubated with primary antibodies at 4 °C overnight. The primary antibodies used were anti-phospho-Akt (Ser473) (1:1000; XP 4060, RRID:AB_2315049); anti-Akt (1:1000; 9272, RRID:AB_329827); anti-phospho-p44/42 MAPK (ERK1/2) (Thr202/Tyr204) (1:1000; 4376, RRID:AB_331772); anti-p44/42 MAPK (ERK1/2) (1:1000; 4695, RRID:AB_390779); anti-phospho-SAPK/JNK (Thr183/Tyr185) (1:1000; 9251, RRID:AB_331659); anti-SAPK/JNK (1:1000; 9252, RRID:AB_2250373); anti-BiP (1:1000; 3177, RRID:AB_2119845), anti-phospho-eIF2α (Ser51) (1:500; 3398, RRID:AB_2230924), anti-CHOP (1:500; 2895, RRID:AB_2089254), anti-phospho-FoxO3a (Ser253) (1:500; 9466, RRID:AB_2106674), anti-NPY (1:1000; 11976, RRID:AB_2716286) were from Cell Signaling technologies (Danvers, MA, USA), anti-NPY2R (1:500; SAB4502029, RRID:AB_10747296) was from Sigma-Aldrich (Saint Louis, MO, USA), anti-phospho-Bad (Ser128) (1:500; BS-0893R, RRID:AB_10859346) was from Bioss Antibodies (Woburn, MA, USA), and anti-β-actin (1:5000; ab6276, RRID:AB_2223210) was from Abcam (Cambridge, UK). After primary antibody incubation, the blot membranes were washed three times with TTBS and incubated with appropriate horseradish peroxidase-conjugated secondary antibodies (anti-rabbit (1:5000; 111–035-003, RRID:AB_2313567) or anti-mouse (1:5000; 115–035-003, RRID:AB_10015289) were from Jackson ImmunoResearch Labs, West Grove, PA, USA) for 1 h at room temperature. The blots were rewashed with TTBS buffer, and the protein bands were detected by enhanced chemiluminescence (Super Signal West Femto Maximum Sensitive Substrate; Thermo Fisher Scientific, Waltham, MA, USA) according to the manufacturer’s instructions. The images were captured using a luminescent image analyzer (ImageQuant LAS 4000, GE Healthcare, Chicago, IL, USA), and the densitometric analysis of the protein band intensities was performed after the relative expression of the target proteins normalized to β-actin using ImageJ software (ImageJ, v 1.52; NIH, Bethesda, MD, USA). 

### 2.5. Immunocytochemistry 

SH-SY5Y cells were seeded at a density of 1 × 10^4^ on gelatin-coated coverslips. After different treatments, the cells were washed with PBS, fixed with 4% paraformaldehyde for 20 min, and washed three times with PBS. Following fixation, the cells were permeabilized with 0.1% Triton X-100 in PBS for 15 min. The cells were washed and blocked with 3% bovine serum albumin in PBS for 1 h at room temperature. Following blocking, the cells were incubated with primary antibodies in antibody dilution buffer (1X PBS/2%, BSA/0.3%, Triton X-100) overnight at 4 °C. The following primary antibodies were used for immunofluorescence staining: anti-phospho-SAPK/JNK (Thr183/Tyr185) (1:500; 9251, RRID:AB_331659); anti-CHOP (1:500; 2895, RRID:AB_2089254), anti-phospho-FoxO3a (Ser253) (1:300; 9466, RRID:AB_2106674); anti-NPY (1:300; 11976, RRID:AB_2716286) and anti-cleaved caspase-3 (Asp175) (1:500; 9664, RRID:AB_2341188) were from Cell Signaling technologies, anti-NPY2R (1:500; SAB4502029, RRID:AB_10747296) was from Sigma-Aldrich (Saint Louis, MO, USA)and anti-phospho-Bad (Ser128) (1:500; BS-0893R, RRID:AB_10859346) from Bioss Antibodies (Woburn, MA, USA). After primary antibody incubation, the cells were washed thrice with PBS and incubated with appropriate secondary antibodies (donkey anti-rabbit Cy3 (1:300; 711–165-152, RRID:AB_2307443) or donkey anti-mouse Alexa Fluor 488 (1:300; 715–545-140, RRID:AB_2340845) were from Jackson ImmunoResearch Labs, West Grove, PA, USA) for 60 min at room temperature. The cells were washed thrice with PBS, and coverslips were mounted with anti-fade mounting media with Prolong DAPI (Life Technologies, Carlsbad, CA, USA). The cells were examined, and images were captured using a Zeiss fluorescence microscope (ZEISS Axio Imager, Carl Zeiss, Oberkochen, Germany).

### 2.6. Statistical Analysis 

All data were graphed and statistically analysed by using GraphPad Prism software (version 8.3.0; GraphPad Software Inc., San Diego, CA, USA). We assessed statistical differences between control and NPY treated group, toxicity (glutamate or tunicamycin) group and toxicity + NPY group, and control and toxicity group by two-way ANOVA group analysis. The significance between these groups were determined by Bonferroni multiple comparison post hoc test. All the results are presented as mean ± standard error of the mean (SEM) from three independent experiments, and a *p*-value of less than 0.05 was considered statistically significant for data analysis.

## 3. Results

### 3.1. NPY Protects SH-SY5Y Cells from Glutamate and Tunicamycin-Induced Cell Death

To investigate the potential protective effect of NPY against glutamate toxicity and tunicamycin stress, we performed an MTT assay to check cell viability. We began the study by understanding glutamate and tunicamycin’s time and concentration-dependent effects on cell viability. Progressive decline in the cell viability was observed at increasing concentrations of glutamate (20 mM to 80 mM) and tunicamycin (1 µg/mL to 8 µg/mL) at 12 and 24 h timepoints (Appendix A). Upon exposure to a 40 mM of glutamate or 1 µg/mL of tunicamycin, around 50% cell death threshold was observed at both time points. Therefore, these concentrations were chosen for further investigations to study the protective effects of NPY against these insults. 

SH-SY5Y cells were exposed to 40 mM of glutamate or 1 µg/mL of tunicamycin and simultaneously treated with different concentrations of NPY (0.1 µM, 0.2 µM, 0.5 µM, and 1 µM). In a control group, the cells were treated with the same concentrations of NPY alone to check the effects of the peptide on cell viability. Compared to control, cells exposed to glutamate showed a significant decrease in cell viability by 1.70 ± 0.04 and 2.06 ± 0.15 at 12 and 24 h (Figure 1B,C). Similarly, cells exposed to tunicamycin also showed a significant decrease in cell viability by 2.50 ± 0.12 and 4.37 ± 0.29 at 12 and 24 h (Figure 1E,F). Glutamate exposed cells which were treated with NPY showed a significant increase (*p* > 0.0001, Figure 1A) in the cell viability against the toxicity. A progressive increase in cell viability was observed with increasing concentrations of NPY, indicating the dose-dependent protective effects of the peptide. NPY treatment was found to exert its most significant protective effect at 1 µM concentration against glutamate toxicity at both the time points. NPY at 1 µM exhibited a significant increase in the cell viability by over 1.8 ± 0.17 and 2.4 ± 0.09 fold at 12 and 24 h, respectively (Figure 1B,C).

Similarly, we found that the NPY treatments exhibited dose-dependent protective effects on cell viability against tunicamycin-induced ER stress (Figure 1D). NPY concentrations at 0.2 µM and 1 µM substantially increased cell viability by over 1.65 ± 0.17 and 2.8 ± 0.3-fold at 12 and 24 h treatments, respectively (Figure 1E,F). There was no significant discernible change in cell viability in cells that solely received NPY treatment. This suggests that NPY treatment alone had no impact on cell death. Furthermore, we immunostained the cells with cleaved caspase-3 antibody to check the cell apoptosis in stress and treatment. We found a higher caspase-3 expression in cells under stress (glutamate and tunicamycin) and a significant decrease in its expression with NPY treatment (Appendix A). These collective results imply that NPY suppressed the glutamate and tunicamycin-induced cell death and corroborated our hypothesis that NPY has a neuroprotective effect against toxic effects of these insults.

### 3.2. NPY Suppresses the Glutamate-Induced Apoptotic Activation of ERK1/2

Previous studies indicate that acute glutamate excitotoxicity induces MAPK hyperactivation and causes ERK1/2-mediated apoptotic death in neurons [41,42,43]. Earlier evidence showed that SH-SY5Y neuroblastoma cells are deficient in NMDA receptors, and CySS-glutamate antiporter mediates glutamate trafficking into the cells [44,45,46]. Thus, we used a higher glutamate concentration (40 mM) to simulate excitotoxicity in these cells. Similar to the MTT assay, SH-SY5Y cells were treated with increasing concentrations of NPY (0.2 µM, 0.5 µM, and 1 µM) to test the protective effect against glutamate (40 mM) at 12 and 24 h of treatments. Immunoblot analysis of these cell lysates revealed that glutamate toxicity induced a sustained activation of ERK1/2 (phosphorylation at Thr202/Tyr204) in SH-SY5Y cells at both times points (Figure 2), and downregulation of its activation was observed with NPY treatments. Compared to ERK1, activation of ERK2 was relatively more prominent with glutamate toxicity and was inhibited in response to NPY treatment. Densitometric quantification showed a significant increase in ERK1/2 activation in glutamate toxicity compared to untreated cells. NPY treatment showed significant inhibitory effects in a dose-dependent manner against glutamate toxicity-induced ERK1/2 activation (*p* < 0.0001; Figure 2B,C,E,F). However, the cells that received NPY treatment alone exhibited a significant dose-dependent activation of both ERK1 and ERK2 in SH-SY5Y cells (*p* < 0.0001; Figure 2). These results suggest that NPY treatment alleviates the glutamate excitotoxicity-induced ERK1/2 activation effects. We further investigated downstream signalling in this MAP kinase cascade to better understand the NPY’s action against glutamate toxicity.

### 3.3. NPY Downregulates the ERK1/2 Mediated Activation of the JNK/BAD Apoptotic Networks in Glutamate Toxicity 

The robust and sustained activation of ERK1/2 by glutamate toxicity is known to induce c-Jun N-terminal kinase (JNK) activation in neuronal cell death [42,43,47]. This activated JNK subsequently catalyses the phosphorylation of Bad at serine 128, thereby promoting Bad-mediated apoptosis [48,49]. To investigate whether NPY-mediated inhibition of ERK1/2 activation in glutamate toxicity regulates JNK/BAD pathway, we subsequently examined the phosphorylation levels of JNK (T183/Y185) and Bad (Ser 128) by Western blot and immunofluorescence analyses (Figure 3 and Figure 4). Immunoblot analysis revealed that SH-SY5Y cells exposed to glutamate (40 mM) resulted in a marked increase in phosphorylation of JNKs at 12 and 24 h of treatment (Figure 3). Densitometric quantification of blots intensities showed a significant increase in JNK1 and JNK2 phosphorylation levels with glutamate toxicity compared to the untreated cells; however, NPY treatments significantly reduced this glutamate-induced activation of JNKs (*p* < 0.0001; Figure 3B,C,E,F). We further investigated the cellular localization of JNKs by immunofluorescence with phospho-JNK (T183/Y185) antibody. In cells exposed to glutamate, we noticed a higher presence of JNK inside the nucleus than in the untreated cells, whereas its nuclear trafficking was decreased with NPY treatment (Figure 3G). The cells treated with NPY alone had no effects on JNK trafficking, indicating the peptide alone has no stress induced effects on the cells. 

We next evaluated the phosphorylation of Bad at serine 128, which is dependent on JNK activation. Phosphorylation of Bad at serine 128 promotes the complex formation with anti-apoptotic Bcl-xL and thus promotes apoptotic pathways [48,50]. Western blot analysis of the cell lysates demonstrated marked upregulation of Bad phosphorylation (Ser 128) under glutamate toxicity conditions, and this upregulation was reduced with dose-dependent NPY treatments (Figure 4). Densitometric quantification demonstrated a significant increase in Bad (Ser 128) phosphorylation followed by its downregulated phosphorylation levels with NPY treatments in glutamate toxicity conditions (*p* < 0.0001; Figure 4B,D). However, no significant increase in phospho-Bad levels was observed in cells treated with NPY alone. We further investigated the treated cells by immunostaining with phospho-Bad (Ser 128) antibody. Similar to the western blot results, a decrease in pro-apoptotic activation of Bad was observed with NPY treatment in immunofluorescence staining of the cells (Figure 4E). These results indicate that the NPY treatment in SH-SY5Y cells induced anti-apoptotic effects through the downregulation of the ERK/JNK cascade as well as Bad phosphorylation in glutamate toxicity conditions. 

### 3.4. NPY Attenuates the ER Stress-Mediated Cell Death Induced by Tunicamycin

To further evaluate the protective action of NPY against tunicamycin-induced cell death, we examined the ER stress markers by Western blot analysis. Tunicamycin is a widely used pharmacological ER stress inducer that promotes ER-stress-mediated cell death over time [32,51]. BiP is a chaperone protein of the ER involved in the degradation of the unfolded proteins and is highly expressed under ER stress conditions [52]. We investigated the activation status of ER stress by analysing markers such as phospho-eIF2α, CHOP, and the expression of BiP in tunicamycin-exposed cells with or without NPY treatments. Western blot analysis revealed substantial increases in the upregulation of BiP, phospho-eIF2α (Ser 51), and CHOP expression caused by tunicamycin, and this upregulation was suppressed markedly with NPY presence at both 12 h and 24 h of treatment. The cells treated with NPY alone did not induce any noticeable changes in BiP and CHOP expression (Figure 5). Quantitative analysis of blot intensities showed that NPY treatment significantly suppressed the tunicamycin-induced upregulation of ER stress markers in a dose-dependent manner at both time points (*p* < 0.01, *p* < 0.0001; Figure 5A–H). The maximum effects of NPY against tunicamycin-induced BiP and CHOP expression were observed at 1 µM at both the time points (*p* < 0.0001; Figure 5B,E,G,H).

Furthermore, we immunostained the cells with CHOP antibody to observe its expression in cells under stress and treatment. In the cells exposed to tunicamycin, we observed a higher CHOP expression inside the nucleus than in the control cells; on the other hand, its nuclear expression was reduced with NPY treatment (Figure 5I). Cells treated with NPY alone showed no significant increase in activation of BiP and CHOP (Figure 5E,G,–I). However, a considerable increase in the phosphorylation of eIF2α (Ser 51) was observed with increasing NPY concentrations (*p* < 0.01, *p* < 0.001; Figure 5C,F). These results together provide the time and dose-dependent protective effect of NPY against tunicamycin-induced ER-stress cell death. 

### 3.5. NPY Alleviates the Oxidative Damage Caused by Glutamate and Tunicamycin in SH-SY5Y Cells

Glutamate excitotoxicity and ER stress have been linked to enhancing reactive oxygen species (ROS) production, causing oxidative damage in neurodegenerative diseases [14,53,54]. This acute oxidative stress promotes the activation of Akt (phosphorylation of Ser 473), which further triggers the phosphorylation of foxhead box O3a (FoxO3a) at sites Ser 253, Thr 32, and Ser 315, leading to its exclusion from the nucleus [55]. This cascade subsequently inhibits the transcriptional activation of FoxO3a, causing oxidative damage in the cells. We sought to delineate the effect of NPY against glutamate and tunicamycin-induced oxidative toxicity by investigating the signalling proteins involved in this pathway. We started by examining the phosphorylation of Akt by Western blotting in cells exposed to glutamate (40 mM) or tunicamycin (1 µg/mL) and treated with NPY for 12 h. The cells exposed to glutamate or tunicamycin exhibited a distinct increase in Akt phosphorylation levels than untreated cells, and this activation was inhibited with NPY treatment. In the cells exposed to glutamate, NPY treatment at a concentration of 1 µM showed a significant decrease in Akt phosphorylation at 12 h (*p* < 0.0001; Figure 6B). Similarly, in tunicamycin-exposed cells, NPY treatment at 0.2 µM, 0.5 µM, and 1 µM concentrations showed a significant decrease in Akt phosphorylation (*p* < 0.0001; Figure 6D). Compared to the untreated cells, a dose-dependent increase in Akt phosphorylation was also evident in cells exposed to NPY alone (*p* < 0.01, *p* < 0.001, *p* < 0.0001; Figure 6B,D). 

To ascertain the oxidative damage in cells by these insults, we further performed a Western blot and immunofluorescence analyses on the same cell conditions by probing with phospho-FoxO3a (Ser 253) antibody. Compared to the untreated cells, increased levels of FoxO3a phosphorylation were noticed in cells exposed to glutamate or tunicamycin (Figure 6E,G). Cells treated with NPY concentrations at 0.2 µM and 0.5 µM showed significant inhibition of FoxO3a phosphorylation under glutamate and tunicamycin stress (*p* < 0.0001; Figure 6F,H). However, a marked increase in FoxO3a phosphorylation was also observed in cells exposed to NPY concentration of 1 µM alone (*p* < 0.01, *p* < 0.001; Figure 6F,H). To access the localization of FoxO3a in response to insults and NPY treatment, we performed an immunofluorescence analysis. We found a higher FoxO3a localization in the subcellular region with insults (glutamate and tunicamycin), whereas, with NPY treatments, increased FoxO3a presence was observed within the nucleus (Figure 6I). On the other hand, at 24 h of treatment, there were no significant changes in Akt and FoxO3a activation. This indicates that the phosphorylation of these proteins was an acute response of cells against oxidative damage caused by glutamate toxicity or tunicamycin stress. These results suggest that NPY treatment decreases the oxidative damage caused by glutamate and tunicamycin in neuronal cells. 

### 3.6. Glutamate Toxicity or Tunicamycin Does Not Affect NPY Receptor (NPY-Y2R) Expression, and NPY Treatment Increases Its Expression in SH-SY5Y Cells 

Among different types of NPY receptors, the neuroblastoma cell line SH-SY5Y was reported to selectively express the NPY-Y2 receptor (NPY-Y2R) [56,57,58,59]. The expression of NPY-Y2R and NPY were examined by Western blot and immunofluorescence analyses under glutamate toxicity, tunicamycin stress, and NPY treatment conditions. The Western blots analysis revealed that the expression of NPY-Y2R was not affected in SH-SY5Y cells either by the insults (glutamate and tunicamycin) or the NPY treatments (Figure 7A,B,E,F). Densitometric quantification of these blots revealed no significant changes in any of these groups (Figure 7C,G). Immunofluorescence analysis of NPY-Y2R expression also revealed no drastic differences in NPY-Y2R expression with treatments (Appendix A). Further, the expression of NPY was found to be absent in untreated cells and cells exposed to glutamate or tunicamycin stress. On the other hand, NPY was found to be expressed in cells only treated with NPY or along with glutamate or tunicamycin stress (Figure 7A,B,E,F). Densitometric quantification of these blots indicated a dose-dependent significant increase in expression of NPY (0.2 µM, 0.5 µM, and 1 µM) in the NPY-treated cells (*p* < 0.01, *p* < 0.001, *p* < 0.0001; Figure 7D,H). Immunofluorescence staining with NPY antibody concurred with the immunoblot results showing higher expression in NPY-treated cells (Figure 7I). These results provide an understanding that the NPY-Y2Rs are well distributed in SH-SY5Y cell and are not affected by stress conditions. Additionally, untreated SH-SY5Y cell lacks or are deficient in NPY, and their presence inside the cell is proportional to the NPY treatment.

## 4. Discussion

Among the multifaceted functions of NPY, its involvement in neurogenesis and neuroprotection has raised it as a potential therapeutic target against neurodegeneration. Over the last few decades, numerous investigations on NPY have demonstrated its therapeutic effectiveness in various neurodegenerative conditions [60,61,62]. However, evidence on underlying molecular mechanisms of NPY that lead to neuroprotection are not well established. This study illustrates the dose and time-dependent action of NPY in downregulating the activation of pro-apoptotic proteins involved in the pathogenesis of neurodegeneration. The insults of glutamate excitotoxicity, ER stress, and oxidative damage are known to be the prominent causes of neurodegeneration [3,63,64]. Using SH-SY5Y neuroblastoma cells as a model, we induced stress to simulate cellular neurodegeneration and co-treated with NPY to investigate its protective effects. 

Glutamate-induced neurotoxicity primarily occurs in two ways; one involves increased stimulation of NMDA receptors which invokes a pathological upsurge in calcium levels leading to cell death [65]. The second type, which is distinct from NMDA-mediated cell death, is mediated by oxidative damage [1]. Despite lacking or deficient NMDA receptor functions in SH-SY5Y cells, glutamate induces toxicity in these cells by oxidative damage via CySS/glutamate antiporters, as reported in earlier studies [33,45]. CySS/glutamate antiporter contributes to the import of cysteine in the cell and glutamate export, which is crucial for regulating the antioxidant glutathione (GSH) level inside the cell [66,67,68]. An imbalance in this transport system can lead to a paradigm called glutamate-induced oxidative toxicity [66]. In this study, we exposed the SH-SY5Y to a higher concentration of glutamate (40 mM) that reverses/destabilize the CySS/glutamate antiporter system, thereby causing depletion of GSH and an increase in ROS production inside the cell as explained in previous studies [69,70]. We hypothesized that NPY’s ability to regulate calcium homeostasis and counteract excitotoxicity could protect the cells from glutamate-induced oxidative toxicity. Hence, we investigated the expression of cell signalling markers involved in glutamate-mediated cytotoxicity. 

The sustained and overactivation of ERK1/2 contributes to the onset of neuronal cell death under glutamate and oxidative stress [43,71,72]. In contrast to ERK1, persistent ERK2 activation is detrimental in causing autophagic or apoptotic stress in neuronal cells [73,74]. Similar to previous findings, our results also showed suppression of ERK1/2 activation under glutamate-oxidative stress conditions, which could protect the cells from apoptosis [75,76]. Our findings revealed an apparent dose-dependent inhibition of ERK1/2 activation by NPY treatment, thereby suppressing the initiation of glutamate-induced apoptosis. It can be inferred that simultaneous NPY treatment with glutamate protects the neuronal cells from apoptotic stress initiation. Interestingly, we noticed a progressive dose-dependent activation of ERK1 and ERK2 in cells treated with NPY alone without glutamate stress. We speculate that NPY treatment as observed in the previous studies could activate ERK/MAPK pathway involved in neuronal proliferation [23,77]. 

To gain a better understanding of the above effects, we further investigated the downstream of the MAPK signalling pathway. Overactivation of ERK alone is insufficient to promote apoptosis in neurons under stress [78,79]; in addition, other MAPK family kinases such as SAPK/JNK and p38 MAPK also play a critical role in apoptotic signalling [78,80]. JNK is a stress-activated kinase involved in both extrinsic and intrinsic apoptotic pathways in neurons, and phosphorylation of its isoforms (JNK1 and JNK2) is required for its full activation. Activated JNKs translocate into the nucleus to phosphorylate the transcription factors (e.g., cJun, Elk3, and ATF2) to promote the expression of pro-apoptotic genes such as TNF-α, Bak, and Fas-L [81,82,83]. Furthermore, JNK can phosphorylate Bad (Bcl2-associated death promoter) protein at Ser 128, liberating it from cytoplasmic sequestration and promoting its interaction with Bcl2 or Bcl-xL in mitochondria [48,84]. This event antagonizes the anti-apoptotic activity of the Bcl2 protein, which further initiates the Bad-mediated apoptotic pathway [85,86,87]. We found that NPY treatment significantly suppressed the pro-apoptotic activation of JNK and Bad phosphorylation under glutamate toxicity (Figure 3 and Figure 4). Our immunofluorescence results clearly showed that nuclear translocation of JNKs under glutamate stress is well reversed with NPY treatment (Figure 3G). These protective effects of NPY were well correlated with the cell viability (Figure 1) and cleaved caspase-3 immunofluorescence results (Appendix A). We also found that NPY treatment alone had no significant effect on the activation of JNK and Bad. These results indicated that the ERK activation by NPY-only treatment is not involved with the apoptotic process. As aforementioned, depending on the type of stress or stimuli, activation of the MAPK pathway has dual opposing effects that either cause cell proliferation or apoptosis [78,88]. Our results suggest that NPY exerts neuroprotective effects against glutamate-induced oxidative toxicity by downregulating the ERK/JNK pathway and apoptotic Bad phosphorylation. 

We further investigated the role of NPY in alleviating ER stress-mediated cell death, which is the most common cause of pathogenesis in many neurodegenerative diseases [3]. The protective action of NPY against ER stress was evaluated by looking at the markers belonging to the PERK signalling pathway. Tunicamycin, at high concentrations, causes a prolonged ER stress condition, eventually leading to apoptotic cell death [89]. Prolonged activation of markers such as PERK, eIF2α, ATF4, and CHOP represents the initiation of ER-stress-mediated cell death [90,91]. Under normal conditions, CHOP (GADD153) is ubiquitously present in the cytosol; and ER stress leads to its induction and accumulation in the nucleus, further downregulating the anti-apoptotic proteins [92,93]. Upon NPY treatment, we found a significant decrease in phospho-eIF2α and CHOP expression, indicating the alleviation of tunicamycin-induced ER stress in SH-SY5Y neuronal cells. Similarly, a time- and dose-dependent decrease in the expression of BiP protein was also noticed with NPY treatment. The expression of BiP protein is enhanced during ER stress to facilitate protein refolding [94]. At 12 h of treatment, a significant decrease in BiP was observed at a higher concentration of NPY treatment (Figure 5). However, an evident downregulation of BiP and CHOP expression was observed at 24 h of treatment, indicating that the NPY treatment relieved the cell from ER stress with time. Likewise, the expression of CHOP inside the nucleus was inhibited by the NPY treatment (Figure 5I). Downregulation of these markers by NPY rids the cell from apoptosis which is ascertained by the cell viability results (Figure 1). Moreover, NPY alone had no significant effect on the activation of these ER markers. The study outcomes established the dose and time-dependent protective effects of NPY against tunicamycin-induced ER stress. These results further strengthen the neuroprotective properties of NPY observed with in vivo and in vitro models of neuronal injury conditions [18,35,62,95,96,97]. 

In addition, we examined the role of Akt and its associated signalling pathway with respect to NPY treatment. Surprisingly, we noticed an activation of Akt in response to acute glutamate and tunicamycin insults. Similar to glutamate-oxidative toxicity, ER stress could also promote ROS generation making neuronal cells more vulnerable to oxidative stress [14,54]. Several studies have observed that ROS overload within the cell could lead to hyperactivation of Akt, which would hasten cell death [98,99,100,101,102]. We hypothesise that this overactivation of Akt could be an acute cell response to oxidative stress. More importantly, hyperactivated Akt consequently inhibits the transcription factors called FOXO proteins. FOXOs are a group of proteins regulating a wide range of cellular functions such as cell differentiation, autophagy, longevity, and DNA repair [103,104]. Amongst all FOXO members, FoxO3a (FKHRL1) is crucial in promoting the expression of antioxidant proteins such as sestrins-3, manganese superoxide dismutase (MnSOD), and catalase [105,106]. The hyperactivation of Akt inhibits FoxO3a by phosphorylating specifically at Ser253, Thr32, and Ser315, which results in the retention of FoxO3a in the cytoplasm [55]. FoxO3a sequestered in the cytoplasm is functionally inactive [104,107]. This event eventually disrupts the expression of antioxidant proteins, leading to ROS-mediated cell death [100]. Our results revealed that NPY treatment reduced the hyperactivation of Akt and phosphorylation of FoxO3a (Ser 253) under glutamate and tunicamycin insults in SH-SY5Y cells (Figure 6). Further, we found that the NPY treatment promoted the translocation of FoxO3a into the nucleus by making it functionally active (Figure 6I). Our findings corroborate previous evidence that NPY activation of FoxO3a could upregulate the antioxidant gene expression, thereby preventing oxidative stress in neurodegenerative conditions [106,107]. 

We found a progressive dose-dependent activation of ERK1/2 and Akt in SH-SY5Y cells in the group treated with NPY alone. Further, NPY alone treated cells showed no significant upregulation of pro-apoptotic pathway mediator proteins such as JNK and Bad. Correspondingly, no substantial change in cell viability was observed in a group with only NPY treatment. These interesting results prompt us to investigate further about the physiological role of NPY and its receptors under normal conditions. Among the different types of NPY receptors, NPY Y1, Y2, and Y5 receptors are reported to be predominantly expressed throughout the mammalian nervous system and exert a multitude of physiological functions, including neuroprotection [95,108,109,110]. Our results suggest that the protective effects of NPY against the apoptotic activation of ERK1/2 and Akt under glutamate toxicity or ER stress could be related to the NPY receptor activation and its function in regulating calcium homeostasis. NPY was previously shown to alleviate glutamate excitotoxicity effects through its receptor via regulating calcium homeostasis, decreasing ROS production, and thereby attenuating oxidative stress-induced apoptosis in hippocampal, cortical, and retinal cells [29,34,56,96,111,112]. Additionally, NPY receptors, via interaction with Gqα or Giβ/γ, can activate mitogen-induced protein kinase/extracellular signal-regulated kinase (MAPK/ERK) and phosphatidylinositol-3-kinase (PI3K)/Akt pathways that are involved in promoting neuroprotection and neurogenesis through cellular differentiation and proliferation processes [30,95,96,113,114]. We also found the NPY-Y2R expression in SH-SY5Y cells was not affected by either glutamate or tunicamycin and NPY treatments [57]. The activation of ERK and Akt with NPY alone treatment in SH-SY5Y cells might occur through NPY-Y2R signalling via Gqα or Giβ/γ that could exert neuroprotection and neurogenesis effects [25,109]. Furthermore, we observed that the NPY expression in SH-SY5Y cells was influenced in response to treatment with this peptide (Figure 7). We speculate that this could be potentially due to trafficking via endocytosis or other means, such as expression changes.

In conclusion, our findings highlighted that NPY attenuates the pro-apoptotic markers involved in glutamate toxicity, ER stress, and oxidative stress in SH-SY5Y cells (Figure 8) and strengthens further the concept of NPY as a potential therapeutic target against neurodegenerative diseases. 

## Figures and Tables

**Figure 1 cells-11-03665-f001:**
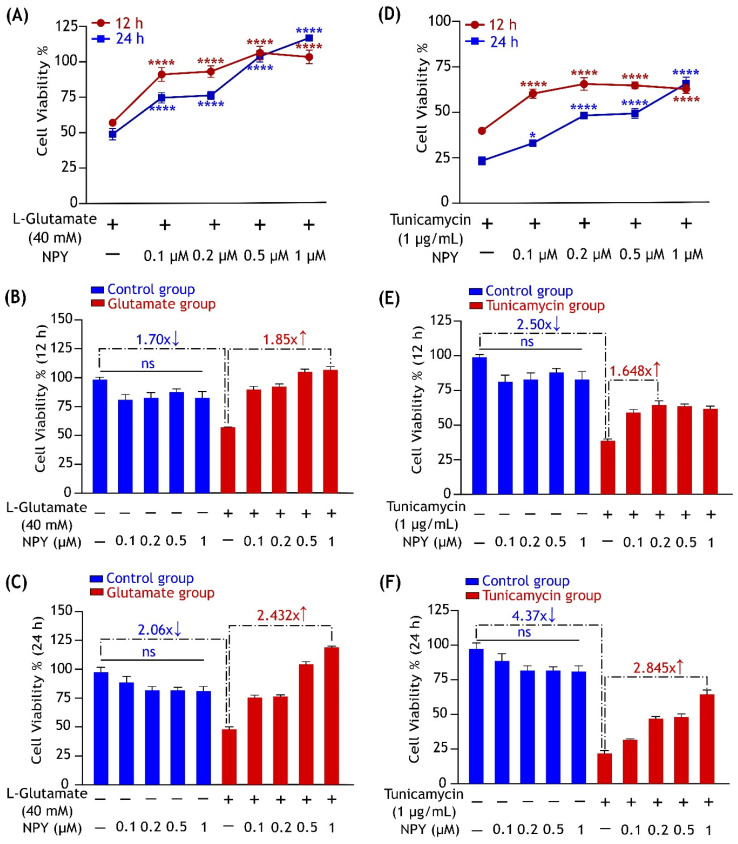
NPY treatment alleviates the cytotoxic effects of glutamate and tunicamycin in SH-SY5Y cell cultures. (**A**) The protective effects of NPY at different concentrations (0.1 µM, 0.2 µM, 0.5 µM, and 1 µM) on cell viability against L-glutamate (40 mM) induced toxicity. Data represent the mean ± SEM of three independent experiments (ns represent non-significant, F (4,50) = 70.15, **** *p* < 0.0001 for 12 h (red), **** *p* < 0.0001 for 24 h (blue), two-way ANOVA followed by Bonferroni’s post hoc test, *n* = 3). Bar graphs represent no significant changes in cell viability after treatment with NPY alone at 12 (**B**) and 24 h (**C**), as well as the fold change difference between the control and glutamate group and between glutamate and glutamate + NPY group showing the highest cell viability. (**D**) The protective effects of NPY at different concentrations (0.1 µM, 0.2 µM, 0.5 µM, and 1 µM) on cell viability against tunicamycin (1 µg/mL) induced cell death. Data represent the mean ± SEM of three independent experiments (ns represent non-significant, F (4,50) = 19.22, **** *p* < 0.0001 for 12 h (red), * *p* < 0.05, **** *p* < 0.0001 for 24 h (blue), two-way ANOVA followed by Bonferroni’s post hoc test, *n* = 3). Bar graphs represent no significant changes in cell viability after treatment with NPY alone (blue) at 12 (**E**) and 24 h (**F**), as well as the fold change difference between the control and tunicamycin group and between tunicamycin and tunicamycin + NPY group showing the highest cell viability.

**Figure 2 cells-11-03665-f002:**
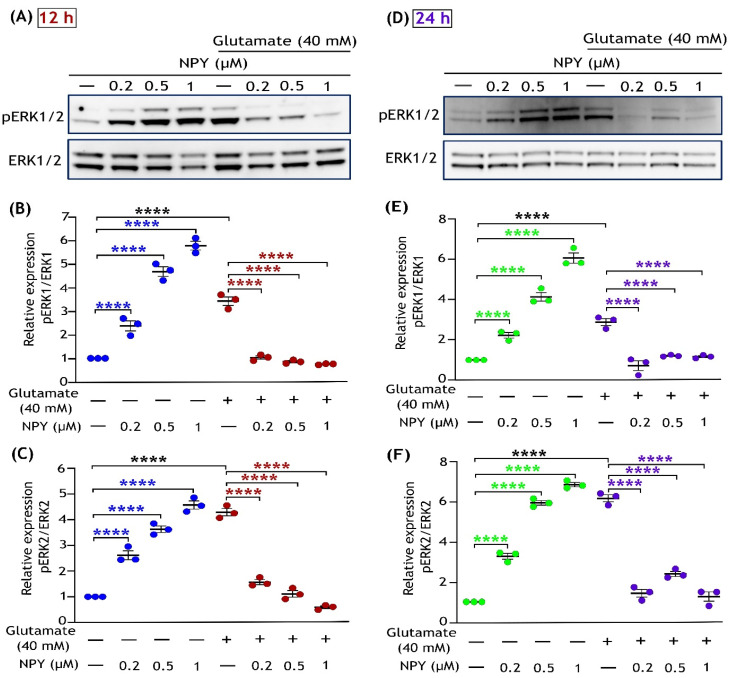
Dose-dependent inhibitory effects of Neuropeptide Y on glutamate-induced activation of ERK1/2 in SH-SY5Y cell cultures. Western immunoblot analysis illustrates the increased phosphorylation of ERK1/2 (T202/Y204) with NPY treatments in the absence of glutamate (control + NPY) at different concentrations, while NPY treatment reduced the glutamate-induced (glutamate + NPY) phosphorylation of ERK1/2 at 12 h treatment (**A**) and densitometric quantification of ERK1 (**B**) and ERK2 (**C**) blot densities indicate a significant increase in ERK1/2 phosphorylation under glutamate treatment compared to control (F(1,16) = 347.7 in Figure (**B**) and F(1,16) = 145.1 in Figure (**C**), **** *p* < 0.0001, *n* = 3) and this activation significantly decreased with increasing concentration of NPY treatment for 12 h (F(3,16) = 29.21 in Figure (**B**) and F(3,16) = 8.319 in Figure (**C**), **** *p* < 0.0001 represents significance between control and NPY group (blue), **** *p* < 0.0001 represents significance between glutamate and glutamate + NPY group (red), *n* = 3). Western immunoblot analysis for 24 h treatment (**D**) and their densitometric quantification of ERK1 (**E**) and ERK2 (**F**) blot densities indicates a significant increase in ERK1/2 phosphorylation under glutamate treatment compared to control (F(1,16) = 254.6 in Figure (**E**) and F(1,16) = 182.3 in Figure (**F**), **** *p* < 0.0001, *n* = 3) and a significant decrease in the glutamate induced phosphorylation of ERK1/2 with increasing concentration of NPY treatment (F(3,16) = 63.96 in Figure (**E**) and F(3,16) = 59.63 in Figure (**F**), **** *p* < 0.0001 represents significance between control and NPY group (green), **** *p* < 0.0001 represents significance between glutamate and glutamate + NPY group (purple), *n* = 3).

**Figure 3 cells-11-03665-f003:**
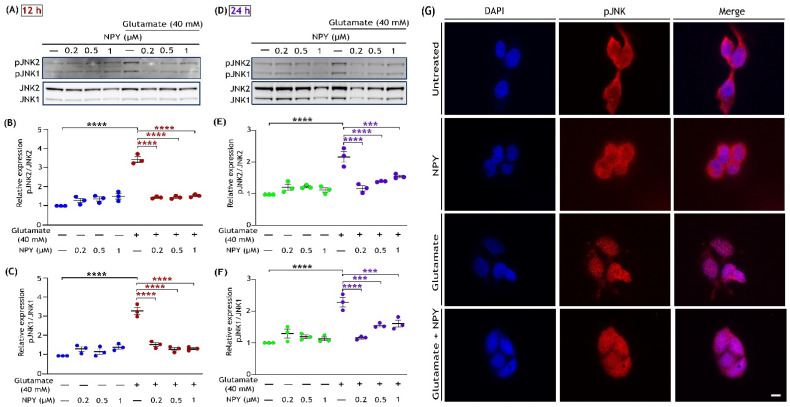
NPY treatment suppresses the activation of pro-apoptotic JNK induced by glutamate in SH-SY5Y cell cultures. Western blot analysis illustrates the glutamate induced phosphorylation of JNK (T183/Y185), while NPY treatments (glutamate + NPY) reduced the phosphorylation of JNK at 12 h (**A**) and densitometric quantification of JNK2 (**B**) and JNK1 (**C**) phosphorylation indicates a significant increase in phosphorylation of JNK induced by glutamate compared to control (F(1,16)= 102 in Figure (**B**) and F(1,16)= 67.85 in Figure (**C**), **** *p* < 0.0001, *n* = 3) and a concentration-dependent decrease with NPY treatments at 12 h (F(3,16) = 34.88 in Figure (**B**) and F(3,16) = 25.90 in Figure (**C**), **** *p*  <  0.0001 represents significance between glutamate and glutamate + NPY group (red), *n* = 3). Western immunoblot analysis for 24 h treatment (**D**) and their densitometric quantification of JNK2 (**E**) and JNK1 (**F**) phosphorylation levels indicates a significant increase in phosphorylation of JNK induced by glutamate compared to control (F(1,16)= 59.13 in Figure (**E**) and F(1,16)= 64.77 in Figure (**F**), **** *p*  <  0.0001, *n* = 3) and a concentration dependent decrease with NPY treatments (F(3,16) = 7.82 in Figure (**E**) and F(3,16) = 7.926 in Figure (**F**), *** *p*  <  0.001 and **** *p*  <  0.0001 represents significance between glutamate and glutamate + NPY group (purple), *n* = 3). (**G**) Immunofluorescence images (representative) of treated SH-SY5Y cells showing the expression of phosphorylated JNKs. After 12 h of treatment, cells were fixed and immunostained with an anti-phospho-JNK (T183/Y185) antibody (red). Nuclei were stained with DAPI (blue). Scale bar = 10 μm.

**Figure 4 cells-11-03665-f004:**
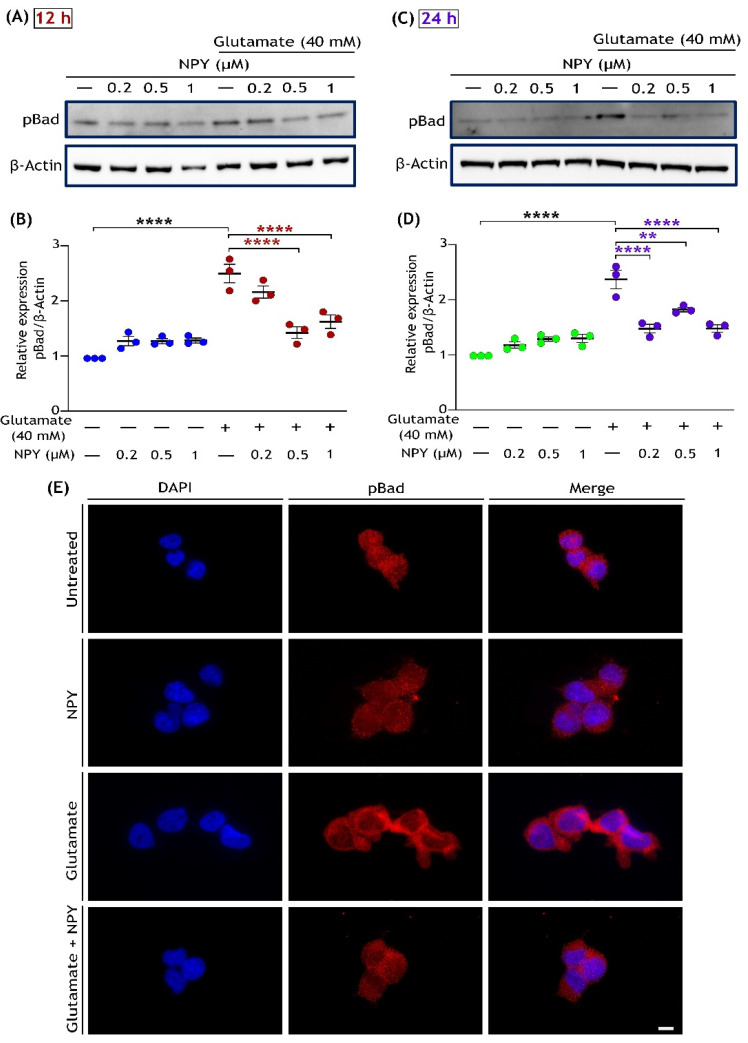
NPY treatment inhibits pro-apoptotic BAD phosphorylation caused by glutamate in SH-SY5Y cell cultures. Western immunoblot analysis for phosphorylation of BAD at Ser 128 under glutamate and NPY treatments (glutamate + NPY) at 12 h (**A**) and the densitometric quantification of phospho-BAD (**B**) reveals a significant increase in BAD activation under glutamate stress compared to control (F(1,16) = 108.1, **** *p*  <  0.0001, *n* = 3) and a decrease with NPY treatment at 12 h (F(3,16) = 7.4, **** *p*  < 0.0001 represents significance between glutamate and glutamate + NPY (red), *n* = 3). Western immunoblot analysis for 24 h treatment (**C**) and the densitometric quantification of phospho-BAD (**D**) reveals a significant increase in BAD activation under glutamate stress compared to control (F(1,16) = 116.1, **** *p*  <  0.0001, *n* = 3), and a decrease with NPY treatments at 24 h (F(3,16) = 7.594, ** *p*  <  0.01 and **** *p*  <  0.0001 represents significance between glutamate and glutamate + NPY (purple), *n* = 3). Each band intensity was normalized to the respective band intensity of β-actin. (**E**) Immunofluorescence images (representative) of treated SH-SY5Y cells showing the expression of phosphorylated Bad. After 12 h of treatment, cells were fixed and immunostained with an anti-phospho-Bad (Ser 128) antibody (red). Nuclei were stained with DAPI (blue). Scale bar = 10 μm.

**Figure 5 cells-11-03665-f005:**
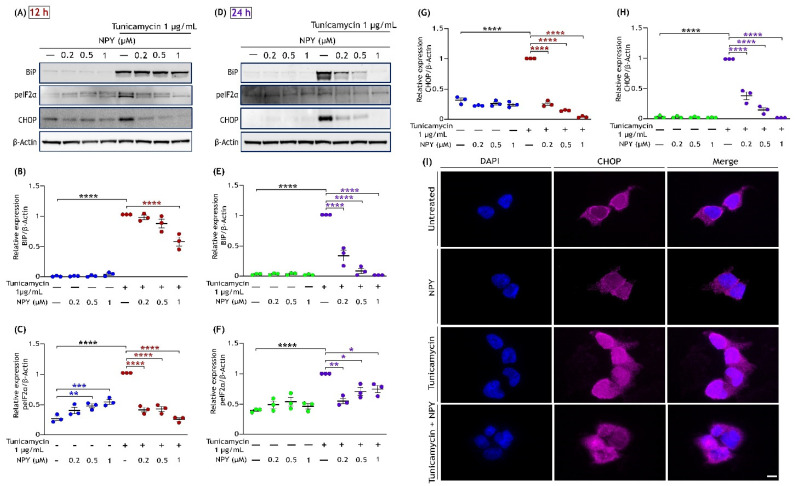
NPY treatment eases the ER stress induced by tunicamycin in SH-SY5Y cells. Western immunoblot illustrates that the tunicamycin-induced activation of ER markers such as BiP, phospho-eIF2α (Ser51), and CHOP that are reduced with the NPY treatment (0.2 µM, 0.5 µM, and 1 µM) at 12 (**A**) and 24 h (**D**). Densitometric quantification of BiP at 12 (**B**) and 24 h (**E**) reveals a significant increase under tunicamycin compared to the control group (F(1,16) = 912.6 in Figure (**B**) and F(1,16)= 181.4 in Figure (**E**), **** *p*  <  0.0001, *n* = 3) and a significant decrease in its expression with NPY treatment (F(3,16) = 10.72, **** *p*  <  0.0001 represents significance between tunicamycin and tunicamycin + NPY in Figure (**C**) (red) and F(3,16)= 85.76, **** *p*  <  0.0001 in Figure (**E**) (purple), *n* = 3). Densitometric quantification of phospho- eIF2α at 12 (**C**) and 24 h (**F**) reveals a significant increase under tunicamycin compared to the control group (F(1,16) = 17.33 in Figure (**C**) and F(1,16)= 34.16 in Figure (**F**), **** *p*  <  0.0001, *n* = 3) and a significant decrease in its activation with NPY treatment (F(3,16) = 20.23, ** *p*  <  0.01, *** *p*  <  0.001 represents significance between control and NPY group (blue), **** *p*  <  0.0001 represents significance between tunicamycin and tunicamycin + NPY in Figure (**E**) (red) and F(3,16)= 5.350, * *p*  <  0.05, ** *p*  <  0.01 in Figure (**F**) (purple), *n* = 3). Densitometric quantification of CHOP at 12 (**G**) and 24 h (**H**) reveals a significant increase under tunicamycin compared to the control group (F(1,16) = 43.85 in Figure (**G**) and F(1,16)= 351.1 in Figure (**H**), **** *p*  <  0.0001, *n* = 3) and a significant decrease in its expression with NPY treatment at 12 h and 24 h, respectively (F(3,16) = 243.6, **** *p*  <  0.0001 represents significance between tunicamycin and tunicamycin + NPY in Figure (**G**) (red) and F(3,16)= 129.5, **** *p*  <  0.0001 in Figure (**H**) (purple), *n* = 3). Each band intensity was normalized to the respective band intensity of β-actin. (**I**) Immunofluorescence images (representative) of treated SH-SY5Y cells showing the expression of CHOP. After 12 h of treatment, cells were fixed and immunostained with an anti-CHOP antibody (pink). Nuclei were stained with DAPI (blue). Scale bar  = 10 μm.

**Figure 6 cells-11-03665-f006:**
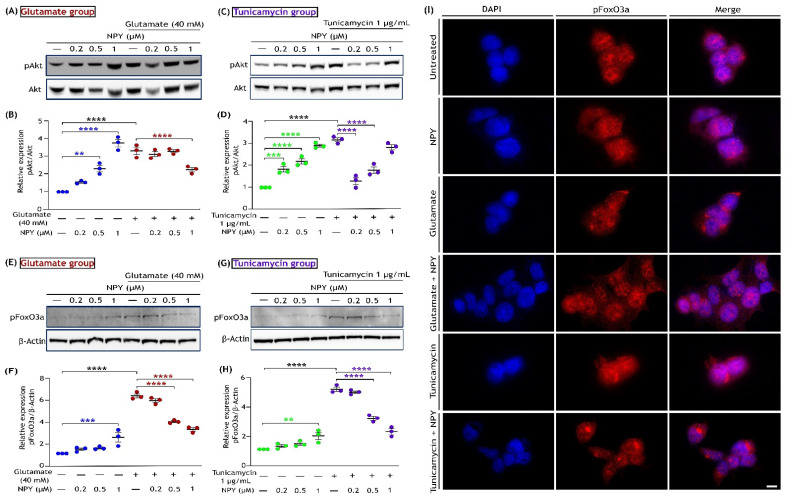
NPY reduces the activation of the Akt/FoxO3a axis triggered by oxidative stress under glutamate and tunicamycin in SH-SY5Y cell cultures. Western immunoblot analysis and densitometric quantification of AKT phosphorylation (Ser473) (**A**,**B**) with glutamate and tunicamycin (**C**,**D**) treatments at 12 h reveal a significant increase in AKT activation with glutamate/tunicamycin compared to the control group (F(1,16) = 69.66 in Figure (**B**) and F(1,16)= 12.93 in Figure (**D**), **** *p*  <  0.0001, *n* = 3) and NPY treatment significantly decreases this activation (F(3,16) = 15.36, ** *p*  <  0.01, **** *p*  <  0.0001 represents significance between control and NPY group (blue), **** *p*  <  0.0001 represents significance between glutamate and glutamate + NPY in Figure (**B**) (red), *n* = 3; F(3,16) = 44.88, *** *p*  <  0.001, **** *p*  <  0.0001 represents significance between control and NPY group (green), **** *p*  <  0.0001 represents significance between tunicamycin and tunicamycin + NPY in Figure (**D**) (purple), *n* = 3). Western immunoblot analysis and densitometric quantification of FoxO3a phosphorylation (Ser 253) (**E**,**F**) with glutamate and tunicamycin (**G**,**H**) treatment at 12 h reveal a significant increase in FoxO3a activation with glutamate/tunicamycin compared to the control group (F(1,16) = 550.6 in Figure (**F**) and F(1,16)= 577.2 in Figure (**H**), **** *p*  <  0.0001, *n* = 3) and NPY treatment significantly decreases this activation (F(3,16) = 12.99, *** *p*  <  0.001 represents significance between control and NPY group (blue), **** *p*  <  0.0001 represents significance between glutamate and glutamate + NPY in Figure (**F**) (red), *n* = 3; F(3,16) = 26.45, ** *p*  <  0.01 represents significance between control and NPY group (green), **** *p*  <  0.0001 represents significance between tunicamycin and tunicamycin + NPY in Figure (**H**) (purple), *n* = 3). Each band intensity of phospho-FOXO3a was normalized to the respective band intensity of β-actin. (**I**) Immunofluorescence images (representative) of treated SH-SY5Y cells showing the expression of phosphorylated FoxO3a. After 12 h of treatment, cells were fixed and immunostained with anti-phospho-FoxO3a (Ser253) antibody (red). Nuclei were stained with DAPI (blue). Scale bar = 10 μm.

**Figure 7 cells-11-03665-f007:**
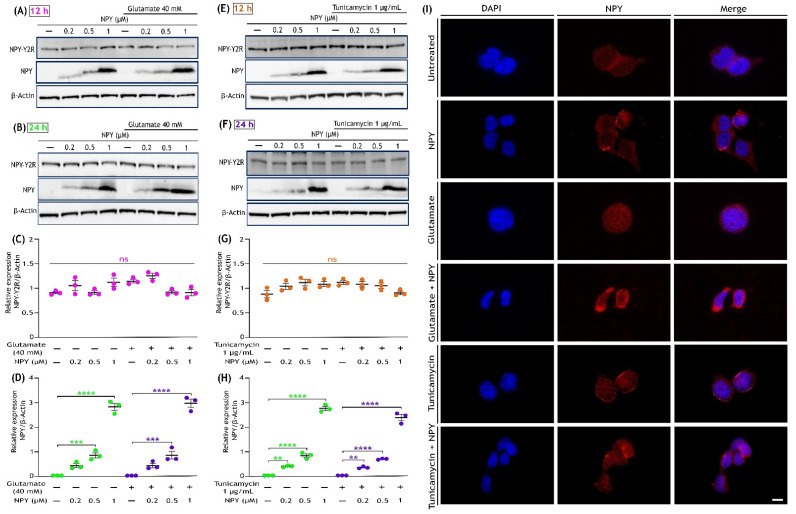
Expression of Neuropeptide Y and NPY-Y2R receptor in SH-SY5Y cells. (**A**,**B**) Western immunoblot analysis illustrates the presence of NPY-Y2R in all indicated groups (Control, Control + NPY, glutamate, and glutamate + NPY) and dose-dependent increase in the NPY expression with NPY treatment at 12 and 24 h, respectively. (**C**,**D**) Densitometric quantification of NPY-Y2R blot intensities represent that its expression was not affected by either the insults (glutamate or tunicamycin) or the NPY treatment at 12 h (F(3,16)= 5.461 in Figure (**C**) and F(3,16) = 1.265 in Figure (**D**), ns represents non-significance, *n* = 3). (**E**,**F**) Western immunoblot analysis illustrates the presence of NPY-Y2R in all indicated groups (Control, Control + NPY, tunicamycin, and tunicamycin + NPY) and dose-dependent increase in the NPY expression with NPY treatment at 12 and 24 h, respectively. (**G**,**H**) Densitometric quantification of NPY blot intensities showed a progressive dose-dependent significant increase in its expression with NPY treatment (F(3,16)= 264.8 in Figure (**G**,**F**) (3,16) = 697.9 in Figure (**H**), ** *p*  <  0.01, *** *p*  <  0.001, **** *p*  <  0.0001 represents significance between control and NPY group (green), ** *p*  <  0.01, *** *p*  <  0.001, **** *p*  <  0.0001 represents significance between insults (glutamate or tunicamycin) and insults + NPY (purple), *n* = 3) and no expression in untreated cells or in cells exposed to only insults (glutamate or tunicamycin). Each band intensity was normalized to the respective band intensity of β-actin. (**I**) Immunofluorescence images (representative) of treated SH-SY5Y cells showing the expression of NPY. After 12 h of treatment, cells were fixed and immunostained with an anti-NPY antibody (red). Nuclei were stained with DAPI (blue). Scale bar = 10 μm.

**Figure 8 cells-11-03665-f008:**
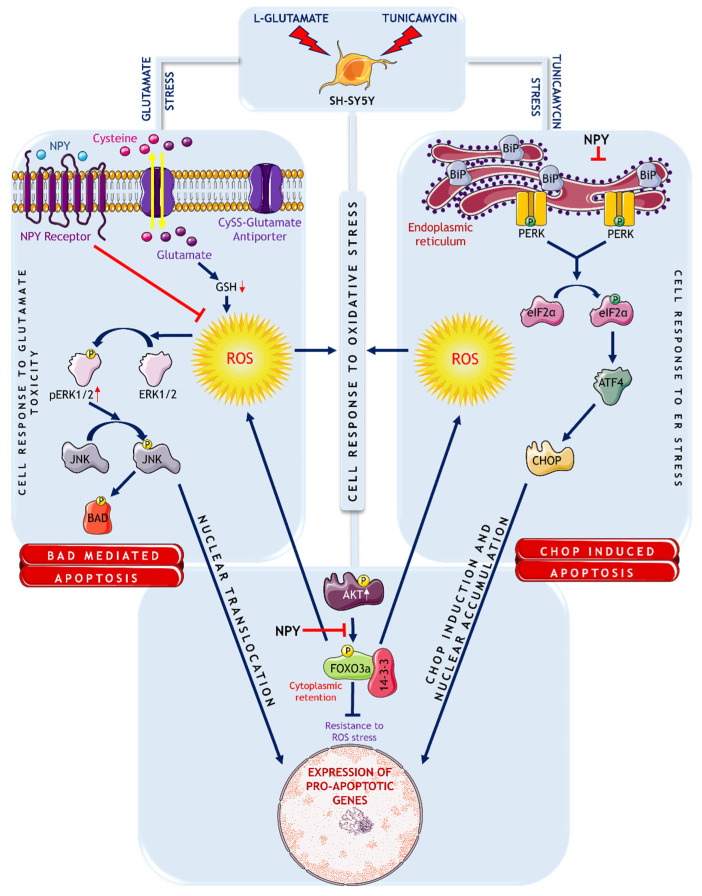
Schematic representation of signalling mechanisms involved in glutamate excitotoxicity and ER stress on SH-SY5Y cells. Source: This figure was created using Servier Medical Art templates, which are licensed under a Creative Commons Attribution 3.0 Unported License; https://smart.servier.com accessed on 30 August 2022.

## Data Availability

Not applicable.

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
