# Peer review of "Neuroprotective Effects of Neuropeptide Y on Human Neuroblastoma SH-SY5Y Cells in Glutamate Excitotoxicity and ER Stress Conditions"

_cells, 2022, doi:10.3390/cells11223665_

Round 1

Reviewer 1 Report

The Article "Neuroprotective Effects of Neuropeptide Y on Human Neuroblastoma SH-SY5Y Cells in Glutamate Excitotoxicity and ER Stress Conditions" by Viswanthram Palanivel et al. describes the protective role of Neuropeptide Y (NPY) against glutamate toxicity and ER stress modelled in vitro by exposure of cells to 40mM glutamate and 1ug/ml tunicamycin respectively. Based on obtained biochemical results, authors conclude that NPY prevents activation of JNK/BAD apoptotic networks in glutamate toxicity via influence on ERK1/2 signalling pathways. Regarding NPY effects on ER stress evoked by tunicamycin, authors show data suggesting that NPY alleviate expression of ER stress markers induced by tunicamycin: p- -eIF2α, CHOP, BiP. Effects of NPY on glutamate Excitotoxicity and ER stress seem to be not related to the activation of the receptor for NPY (NPY-Y2R). The scientific approach presented in the manuscript is interesting, and the results are original. My critical remarks mainly concern the statistical description of obtained results and methodological elucidations, which are low and significantly reduce scientific sounds presented in the manuscript data. Therefore, my recommendation is for the manuscript to go under 

MAJOR REVISION. 

My specific remarks on the manuscript:

  1. Please add references to your statement in lines55-56 about apoptosis promotion when the UPR system is disturbed.
  2. What is the rationale for the range of dose selection for glutamate, tunicamycin, and Neuropeptide Y? Please add an explanation in the 2.2 chapter. Also, I need help finding in the material and method section the rationale of tunicamycin usage to model ER stress in vitro - please complete it. 
  3. Once entered, the abbreviation should be used consistently. Some abbreviations and full names are misused in the article. e.g., please move the explanation for DMSO from line 126 to line 105 (when the first time was used); line 64 - remove the "central" word and brackets in "CNS" - CNS abbreviation is introduced in 43; please be consistent in abbreviations usage – did you mean: "NPY-Rs" (like in line 72) or "NPYRs (without a hyphen, like in line 73) or rather without abbreviations "NPY receptor" (like in line 82)? Please use it without additional explanations later on in the manuscript. Please also check other abbreviations usage and improve the manuscript.
  4. Lines 141-142 - Please clarify how you measured the total protein concentration you isolated. I noted you should have mentioned the type of standard, method of detection, and calculation. Since the total protein concentration has been quantified, why did you use "about 10 ug", not precise 10 ug?
  5. Please complete and correct, if necessary, the description of the statistical analysis you performed. The statement in lines 194-196 needs to be clarified because readers need help understanding the rationale of using two-way ANOVA followed by t-tests with Bonferroni adjustment for your data. First, explain factors for two-way ANOVA and the types of "multiple comparisons" you performed. Explain also why you did not use post hoc tests dedicated to ANOVA.
  6. Statistical reports in the Results section/Figure captions/Supplementary Figure captions are incomplete - Please provide an appropriate report of statistics for your test (F for ANOVA), accompanied with degrees of freedom (df) for the results you describe. Additionally, for indicated "p-value" in a posthoc analysis, please add reference group (vs which group you showed significance with one, two, three, four stars)
  7. Figure captions need to be made easier to understand. Please make consistent figure captions with the figure layout. I found some inconsistencies, e.g. in Fig 1 descriptions with figure layout. From the description of Fig. 1, it is not clear the difference in the effects showed in A from C and D effects and what exactly mean stars indicated in A and B. Similarly, in Figures 2-7, it is not clear what exactly mean black, blue, red, green stars (nowhere reference group is indicated). Also, in the reviewer's opinion, it would be easier to read if effects for glutamate toxicity presented in the left column were numbered first (e.g. for Fig 1 A-C) and results for tunicamycin evoked by ER stress were numbered later on (e.g. for Fig 1 D-F).
  8. In the original Western Blot images, you present them as a separate file (to prove the good quality of collected data), but you need to indicate which band among many unspecific bands was analyzed. It looks like you should have used the molecular mass standard. What do you mean by the line "untreated"? Please indicate clearly which band was analyzed by you and describe its molecular mass. Explain how you assessed the specificity of studied proteins. If "untreated" means >molecular mass marker, please describe the molecular weight of particular marker bands.
  9. In original images, only representative blots n=1 are presented, while in the manuscript author stated they performed n=3. Please send to Editor all analyzed blots or indicate clearly in the "Data Availability Statement" (line 655) where the readers can find all blots which support the data presented in the manuscript data.
  10. Did you verify in other ways than "viability test" glutamate-induced toxicity and tunicamycin-induced ER stress? Did you check, e.g. nuclear condensation, ROS level, and the level of any pro-apoptotic genes after glutamate and tunicamycin? It would be beneficial to show the presence of apoptosis in your in vitro model. 

Author Response

Thanks for the critical comments. Please see the attachment.

Reviewer 2 Report

  The manuscript “Neuroprotective Effects of Neuropeptide Y on Human Neuro-2 blastoma SH-SY5Y Cells in Glutamate Excitotoxicity and ER 3 Stress Conditions” discusses the protective role of neurotransmitter NPY in glutamate and Tunicamycin induced toxicity and how it could be a relevant therapeutic intervention against oxidative stress induced cell death. A few queries mentioned below, if answered, can make the overall study more attractive and relevant:

1.       The supplementary figure for MTT assay shows around 50% viability with 1μg/ml tunicamycin, however in figure 1B the first bar with same dose of tunicamycin without NPY shows a much-reduced viability, indicating a high toxicity of this dose which isn’t considerably restored with NPY doses (close to just 60% with the highest dose of NPY at both time points). Please explain.

2.       Were the 2 sets (control and glutamate; control and tunicamycin) in figures 1C-F compared to each other statistically? If not, what is the relevance of these figures since the sole objective of the experiment is to ascertain if addition of toxin changes anything in the viability of NPY exposed cells or vice-versa? Also how does reduced cell viability in tunicamycin just indicate ER stress as mentioned in the legend?

3.       In figure 2A-B the authors show increased phosphorylation of ERK1/2 by NPY at different concentrations and also report decrease in this phosphorylation level by NPY in presence of glutamate compared to glutamate alone group. In the discussion part they address this as proliferative effect of NPY alone and a stress inducing effect in presence of glutamate. Can they explain this result more specifically since it’s more of a speculation than a fact? Additional experiments or maybe just western blots for relevant proteins can further prove this claim.

4.       High quality images for figure 3A (blot for p-JNK), 3G, 4E, 5I, 6I and 7I are required.

5.       What was the objective behind measuring NPY levels after treating the cells with NPY itself?

Author Response

(The authors gave the same response as above.)

Round 2

Reviewer 1 Report

The authors addressed all my critical comments and applied necessary corrections to improve descriptions of presented data. The corrected version of the manuscript is in my opinion ready for publication.